# An Improved Protocol for Targeted Differentiation of Primed Human Induced Pluripotent Stem Cells into HLA-G-Expressing Trophoblasts to Enable the Modeling of Placenta-Related Disorders

**DOI:** 10.3390/cells12162070

**Published:** 2023-08-15

**Authors:** Ian O. Shum, Sylvia Merkert, Svitlana Malysheva, Kirsten Jahn, Nico Lachmann, Murielle Verboom, Helge Frieling, Michael Hallensleben, Ulrich Martin

**Affiliations:** 1Leibniz Research Laboratories for Biotechnology and Artificial Organs (LEBAO), Department of Cardiothoracic, Transplantation and Vascular Surgery (HTTG), Hannover Medical School, 30625 Hannover, Germany; 2REBIRTH-Research Center for Translational and Regenerative Medicine, Hannover Medical School, 30625 Hannover, Germany; 3Biomedical Research in Endstage and Obstructive Lung Disease (BREATH), Member of the German Center for Lung Research (DZL), Hannover Medical School, 30625 Hannover, Germany; 4Laboratory of Molecular Neurosciences, Department of Psychiatry, Social Psychiatry and Psychotherapy, Hannover Medical School, 30625 Hannover, Germany; 5Department of Pediatric Pneumology, Allergology and Neonatology, Hannover Medical School, 30625 Hannover, Germany; 6Institute of Transfusion Medicine and Transplant Engineering, Hannover Medical School, 30625 Hannover, Germany

**Keywords:** HLA-G, trophoblast, extravillous trophoblast cells, primed hiPSC

## Abstract

Abnormalities at any stage of trophoblast development may result in pregnancy-related complications. Many of these adverse outcomes are discovered later in pregnancy, but the underlying pathomechanisms are constituted during the first trimester. Acquiring developmentally relevant material to elucidate the disease mechanisms is difficult. Human pluripotent stem cell (hPSC) technology can provide a renewable source of relevant cells. BMP4, A83-01, and PD173074 (BAP) treatment drives trophoblast commitment of hPSCs toward syncytiotrophoblast (STB), but lacks extravillous trophoblast (EVT) cells. EVTs mediate key functions during placentation, remodeling of uterine spiral arteries, and maintenance of immunological tolerance. We optimized the protocol for a more efficient generation of HLA-G^pos^ EVT-like trophoblasts from primed hiPSCs. Increasing the concentrations of A83-01 and PD173074, while decreasing bulk cell density resulted in an increase in HLA-G of up to 71%. Gene expression profiling supports the advancements of our treatment regarding the generation of trophoblast cells. The reported differentiation protocol will allow for an on-demand access to human trophoblast cells enriched for HLA-G^pos^ EVT-like cells, allowing for the elucidation of placenta-related disorders and investigating the immunological tolerance toward the fetus, overcoming the difficulties in obtaining primary EVTs without the need for a complex differentiation pathway via naïve pluripotent or trophoblast stem cells.

## 1. Introduction

The outer layer of the blastocyst, the trophectoderm, is the precursor to all trophoblast lineages in the placenta. Trophoblast cells are divided into three main subtypes: villous cytotrophoblasts (CTBs), syncytiotrophoblasts (STBs), and extravillous trophoblasts (EVTs), each having a specialized role that is instrumental for the continual growth and success of the developing embryo. Abnormalities at any stage of trophoblast development may therefore result in pregnancy-related complications, namely preeclampsia, fetal growth restriction, or miscarriages [1,2,3]. Many of these adverse outcomes are discovered later in pregnancy, but the underlying pathomechanisms are constituted during the first trimester [4]. Furthermore, placental deficiencies not only affect the immediate gestational period, but also postnatal life [5]. It has also been shown that certain embryonic defects can be rescued once the gene function is restored in the placenta [6,7,8], indicating the importance of understanding placental development not just in the context of pregnancy, but also in many diseases later in life. However, acquiring developmentally relevant tissue material to elucidate placenta-related disease mechanisms is difficult. For instance, it is challenging to get access to HLA-G^pos^ human EVTs, especially from first-trimester placenta. HLA-G^pos^ EVT cells are extremely valuable, not only to model different developmental diseases, but especially for understanding mechanisms of the peripheral immune tolerance of the mother toward her placenta and fetus. This lack of access to primary human placental cells has led to the development of different animal models and the use of human trophoblast cell lines. Even though these model systems have led to new insights into human disease, they cannot fully recapitulate the characteristics of the human phenotype [9,10].

Human pluripotent stem cell (hPSC) technology can alleviate restricted access and provide a renewable source of relevant cells. The initial finding of Xu et al. [11], demonstrating bone morphogenic protein 4 (BMP4)-induction of trophoblasts from human embryonic stem cells (hESCs) was surprising and controversial for many years, since it was commonly believed that only totipotent cells, up to the morula stage, would be able to generate the trophoectoderm. The observation that BMP4 treatment of hPSC also resulted in the expression of mesodermal and endodermal cell markers further casted doubts in the ability of hPSCs to generate trophoblasts [12]. However, Amita et al. were able to improve the robustness of BMP4 trophoblast-like cell generation by inhibiting the induction of mesoderm and endoderm lineage differentiation via two small molecules: A83-01 (activin/NODAL/TGF-β pathway inhibitor) and PD173074 (FGF/VEGF receptor inhibitor). With that, it has been confirmed that hPSCs, including both human embryonic and induced pluripotent stem cells (hiPSCs), are able to differentiate into cells of the trophoblast lineage [13,14]. The combination of BMP4, A83-01, and PD173074 (BAP) treatment drives trophoblast commitment of hPSCs, with characteristic markers for all three main trophoblast subtypes detectable within resultant cultures. Strikingly, transcriptome analysis of specific trophoblast cells in culture via the current BAP differentiation protocol revealed that the cultures are mainly composed of cells that resemble primary STBs [15,16], but apparently do not contain considerable numbers of CTB or EVT-like cells. CTBs are the precursors of both STBs and EVTs, by either fusing together or migrating out of the placenta, respectively. Even though HLA-G^pos^ EVTs play a key role in placental development such as uterine spiral artery remodeling, and especially in immunomodulation of the maternal immune system [17,18,19,20], they are still barely understood. However, it is obvious that HLA-G has a key role in establishing fetal–maternal tolerance during early pregnancy [21].

The generated hPSC-derived trophoblast cultures express cytokeratin 7 (KRT7), with a minority of cells co-localizing with GATA-binding protein 3 (GATA3) and TF activator protein-2 gamma (TFAP2C). Other markers in these hiPSC derivatives were typically not characterized thoroughly, particularly according to the criteria for primary first-trimester mononuclear trophoblast cells, as established by Lee et al., which include their characteristic expression of HLA molecules [22]. These criteria comprise the following: (i) triple presence of the protein markers KRT7, GATA3, and TFAP2C; (ii) lack of all HLA class I molecules for CTB and STB, the absence of HLA-A and -B, and the presence of HLA-G for EVT; (iii) hypo-methylation of the *ELF5* promoter region; and (iv) the expression of four chromosome 19 micro-cluster (C19MC) microRNAs (miRNAs) at similar levels to either choriocarcinoma cell lines Jeg3 or JAR.

Hence, there is a clear need to characterize differentiated hPSC-derived trophoblast cell cultures more thoroughly. Finally, complex in vitro models are needed to explore different aspects of human placenta function during pregnancy, the role of trophoblasts subtypes, and pathomechanisms of placental dysfunction. This will require improved protocols for targeted differentiation into individual trophoblast subtypes showing characteristics of their primary placenta-derived counterparts, including an efficient formation of CTBs and especially EVTs.

In this study, we have focused on the development of a culture protocol only using a chemically defined medium that stimulates the differentiation of primed hiPSCs into EVTs, which express HLA-G on their surface, one of the most striking characteristics of placental EVTs [23]. Applying the criteria developed by Lee et al. [22] for the characterization of primary trophoblast lineages, we demonstrate that our improved chemically defined medium generates substantially higher amounts of EVT cells than previously published protocols, and also yields multinucleated trophoblast giant cells (MTGC), another cell type present in the placenta.

## 2. Materials and Methods

### 2.1. hiPSC Culture

The human iPSC lines Isis (MHHi007-A) [24] and Osiris were generated by reprogramming CD34^pos^ cells or HFF-1 cells (ATCC number: SCRC-1041), respectively, using the Cytotune^®^-iPS Sendai Reprogramming Kit (Thermo Fisher Scientific, Waltham, MA, USA). The human iPSC line Moo-2 was generated from CD34^pos^ cells and lentiviral overexpression of *OCT4*, *SOX2*, *KLF4*, and *c-MYC*. For the study, all iPSC lines were maintained under feeder-free conditions, on Geltrex^TM^ (Gibco™, A1413202, Billings, MT, USA)- with an in-house-produced Essential 8 (E8) medium (DMEM/F-12 (Gibco™, 11330057, Billings, MT, USA) supplemented to 543 mg/L sodium bicarbonate (Merck, S5761, Darmstadt, Germany), 20 mg/L insulin (Merck, I9278, Darmstadt, Germany), 64 mg/L ascorbic acid 2-phosphate (Merck, A8960, Darmstadt, Germany), 14 µg/L sodium selenite (Merck, S5261, Darmstadt, Germany), 10.7 mg/L human recombinant Transferrin (Merck, T3705, Darmstadt, Germany) 100 µg/L bFGF (Peprotech, 100-18B, Hamburg, Germany), 2 µg/L TGF-β1 (Peprotech, 100-21C, Hamburg, Germany)). The medium was changed daily, and cells were passaged as single cells using Accutase^®^ (Pan Biotech, P10-21500, Aidenbach, Germany) when surface area reached over 80%.

### 2.2. Differentiation into Trophoblast-Like and EVT-like Cells

Differentiation of iPSC into BAP trophoblast-like cells was performed as following: iPSC were seeded at a density of 4000 cells/cm^2^ into either a 6-well or 12-well plate coated with Geltrex^TM^ and cultured in 2 mL E8 + RI medium (E8 supplemented to 10 mM Y-27632 (Tocris, 1254, Wiesbaden-Nordenstadt, Germany)). After 2 days, medium was replaced with 2 mL E6-BAP (E8 medium minus bFGF and TGF-β1, supplemented to 10 ng/mL BMP4 (RnD Systems, 314-BP, Abingdon, UK), 1 µM A83-01 (Tocris, 2939, Wiesbaden-Nordenstadt, Germany), and 0.1 µM PD173074 (Sigma Aldrich, P2499, Saint Louis, MO, USA). The medium was replaced every 2 days and the cells were analyzed at the specified time points. For the differentiation of BhAhP EVT-like cells, iPSCs were seeded at a density between 1000 and 3000 cells/cm^2^ (cell line-dependent) either on a 6-well or 12-well plate, coated with Geltrex^TM^ and cultured in 2.75 mL or 1 mL (6-well and 12-well plate, respectively) E8 + RI medium. After 2 days, the medium was replaced with E6-BAP, but with 0.4 µM PD173074. After 24 h, the medium was replaced with BhAhP (10 ng/mL BMP4, 7.5 µM A83-01, 0.4 µM PD173074). Medium was replaced every second day.

### 2.3. Differentiation into Early Mesoderm and Endoderm Cells

To differentiate into early mesoderm, we adapted a previously published method [25]. hiPSCs were dissociated into single cells and seeded to aggregate at 0.5 × 10^6^ cells/mL in 3 mL of E8 + RI in 6-well suspensions. After 24 h, the medium was exchanged to E8. On day 3, aggregates were reseeded at 0.5 × 10^6^ cells/mL in 3 mL into 6-well suspensions, where differentiation was initiated with CDM3 (RPMI1640 (with 2 mM glutamine) supplemented with 495 µg/mL human recombinant albumin (ScienCell, SC-OsrHSA, Carlsbad, CA, USA) and 213 µg/mL ascorbic acid + 7.5 µM CHIR99021 (Leibniz University Hannover, Hannover, Germany). The cells were harvested 24 h after differentiation initiation.

To differentiate into the early endoderm, iPSCs were differentiated using the STEMdiff definitive endoderm kit (STEMCELL Technologies, 05110, Vancouver, BC, Canada), according to the manufacturer protocol.

### 2.4. Flow Cytometry Analysis

Differentiated cells were dissociated with TrypLE^TM^ (Thermo Fisher Scientific, 12604013, Waltham, MA, USA) for 13 min at 37 °C. The cells were then gently dislodged and pipetted three times and centrifuged for 3 min at 300× *g* at 4 °C. Then, the cells were resuspended in a 1% BSA solution. For each sample, 100,000 cells in 100 µL were used. Primary and secondary antibodies were incubated for 30 min at 4 °C in the dark, with three washes of PBS in between. The cells were resuspended in 50 µL of a 0.66 µg/mL DAPI solution to detect dead cells. Data were acquired via a MACSQuant Analyzer 10 (Miltenyi Biotec, Bergisch Gladbach, Germany), and analyzed using FlowJo V10 (Tree Star, Ashland, OR, USA).

All antibodies used in flow cytometry are listed in Appendix A.

### 2.5. Immunofluorescence Staining

Immunofluorescence staining of the differentiated cells was performed after paraformaldehyde fixation or as live cultures. The cells were fixed with 4% paraformaldehyde for 15 min at room temperature. For immunostaining, a 10% normal donkey serum + 0.3% Triton X-100 (Sigma Aldrich) PBS solution was used for blocking for 20 min at room temperature. Primary and secondary antibodies were diluted in 300 µL of 0.1% BSA + 0.3% Triton X-100 PBS solution. Incubations were performed on an orbital shaker for either 1 h at room temperature or overnight at 4 °C for primary antibodies, while secondary antibodies were incubated for 30 min at room temperature. After each incubation, the cells were washed three times with PBS. For live-cell immunostaining, primary and secondary antibody incubation was performed at 37 °C 5% CO_2_ for 30 min. Before each incubation, the cells were washed once with the cell culture medium. Immunostained cells were fixed with 4% paraformaldehyde for 15 min at room temperature. Microscopy was performed on a Zeiss Observer A1 or A7. All antibodies used for immunofluorescence staining are listed in Appendix A.

### 2.6. Methylation Analysis

Genomic DNA was extracted from snap-frozen pellets according to the “QIAamp^®^ DNA blood kit” (QIAGEN, 51106, Hilden, Germany). Of the genomic DNA, 500 ng was prepared in accordance with the “EpiTect 96 Bisulfite Kit protocol” (QIAGEN, 59104, Hilden, Germany). The thermal cycler was set according to the manufacturers’ protocol. The amplification PCR (primers 1219 and 1225, cycler program Amp PCR (97 °C, 15 min, 97 °C, 1 min, (95 °C, 30 s, 68 °C, 45 s, 68 °C, 1 min) 15× cycles, incremental −1 °C/cycle, (95 °C, 30 s, 53 °C, 30 s, 68 °C, 45 s) 20× cycles, 65 °C, 5 min, 12 °C, hold)) was followed by automated purification on the Biomek NX^P^ Automated Workstation (Beckman Coulter, Inc., Brea, CA, USA) through the use of paramagnetic beads (CleanNGS, GC biotech, CNGS-0050, Waddinxveen, The Netherlands). The sequencing PCR was prepared using the “BigDye™ Sequencing Kit” (Applied Biosystems™, 4337458, Waltham, MA, USA), and the product was purified using paramagnetic beads (ClearDTR, GC Biotech, CDTR-0005, Waddinxveen, The Netherlands) on a Biomek NX^P^ Automated Workstation. To ensure the sequencing of all CpGs in the region of interest, the sequencing PCR was performed with two approaches: 1. Forward primer 1221 and cycler protocol SeqATRev (95 °C, 1 min, (96 °C, 5 s, 60 °C, 90 s, 50 °C, 90 s), 25× cycles, 12 °C, hold); 2. Reverse primer 1225 and cycler protocol SeqSTD (96 °C, 1 min, (96 °C, 10 s, 50 °C, 5 s, 60 °C, 4 min), 28× cycles, 12 °C, hold). Sequencing was performed on a Sanger 3500xL Genetic Analyzer (Applied Biosystems™). All samples were measured in triplicates. Primers can be found in Appendix A.

### 2.7. RNA Isolation and miRNA qRT-PCR

Samples were lysed in TRIzol (Life Technologies, 15596-026, Waltham, MA, USA), according to the manufacturers’ protocol. Total RNA was purified using the “NucleoSpin RNA kit” (Macherey-Nagel, 740955, Nordrhein-Westfalen, Germany), according to the manufacturers’ protocol. To quantify C19MC miRNAs, we adapted the method described by Lee et al. [22]. Of the RNA, 500 ng was converted to cDNA using the “TaqMan MicroRNA reverse transcription kit” (Applied Biosystems™, 4366596, Waltham, MA, USA). The reaction volume was 15 µL with 50 nM of RT primer. Quantification via qRT-PCR was performed with “SsoAdvanced Universal SYBR^®^ Green Supermix” (Bio-Rad, 1725274, Hercules, CA, USA). Primers can be found in Appendix A.

### 2.8. Library Generation

Of the total RNA, 500 ng per sample was utilized as the input for the mRNA enrichment procedure with “NEBNext^®^ Poly(A) mRNA Magnetic Isolation Module” (New England Biolabs, E7490L, Ipswich, MA, USA), followed by stranded cDNA library generation using “NEBNext^®^ Ultra II Directional RNA Library Prep Kit for Illumina” (New England Biolabs, E7760L, Ipswich, MA, USA). All steps were performed as per recommended by the manufacturer, except that all reactions were downscaled to 2/3 of the initial volumes. Furthermore, one additional purification step was introduced at the end of the standard procedure, using 1× “Agencourt^®^ AMPure^®^ XP Beads” (Beckman Coulter, Inc., A63881 Brea, CA, USA). cDNA libraries were barcoded via a dual indexing approach, using “NEBNext Multiplex Oligos for Illumina, 96 Unique Dual Index Primer Pairs” (New England Biolabs, 6440S, Ipswich, MA, USA). All generated cDNA libraries were amplified with seven cycles of the final PCR. The fragment length distribution of individual libraries was monitored using “Bioanalyzer High Sensitivity DNA Assay” (Agilent Technologies, 5067-4626, Santa Clara, CA, USA). Quantification of the libraries was performed through use of the “Qubit^®^ dsDNA HS Assay Kit” (Thermo Fisher Scientific, Q32854, Waltham, MA, USA).

### 2.9. Library Denaturation and Sequencing Run

Equal molar amounts of eight individually barcoded libraries were pooled for a common sequencing run. The library pools were denatured with NaOH, and were finally diluted to 2 pM according manufacturer recommendation. Of each denature pool, 1.3 mL was loaded on an Illumina NextSeq 550 sequencer using a High-Output Flowcell for single reads (Illumina, 20024906, San Diego, CA, USA). Sequencing was performed with the following settings: sequence reads 1 and 2 with 38 bases each; index reads 1 and 2 with 8 bases each.

### 2.10. Raw Data Processing and Bioinformatics Analysis

BCL files were converted to FASTQ files using the bcl2fastq Conversion Software version v2.20.0.422 (Illumina, San Diego, CA, USA). Raw data processing was conducted through the use of nfcore/rnaseq (v1.4.2). The genome reference and annotation data were taken from GENCODE.org (Homo sapiens; GRCh38.p13; release 34). Normalization and differential expression analysis were performed on Galaxy (v20.05) at the RCU Genomics, (Hannover Medical School, Hannover, Germany) with DESeq2 (Galaxy Tool v2.11.40.6) [26] with default settings, except for “Output normalized counts table”, which was set to “Yes”, and all additional filters were disabled (“Turn off outliers replacement”, “Turn off outliers filtering”, and “Turn off independent filtering” set “Yes”).

## 3. Results

### 3.1. Optimization of BAP Treatment Allows for a More Efficient Generation of HLA-G^pos^ EVT-Like Trophoblasts from Primed hiPSCs

Three hiPSC lines were maintained with the chemically defined medium E8 [27]. Similar to Wei et al. [14], trophoblast differentiation was performed in the E6 medium with BAP (BMP4, A83-01, and PD173074) for 9 days (Figure 1A). In line with previous reports [13,14], BAP treatment rapidly changed the morphology of pluripotent stem cells toward a flattened trophoblast-like epithelium. This morphology change coincided with the presence of CDX2 in the nucleus in almost all cells (Figure 1B). In comparison to hiPSC differentiating into the early mesoderm and early endoderm, BAP treatment led to a negligible up-regulation of mesendodermal markers, *TBXT*, *FOXA2*, and *SOX17* (Figure 1C). This indicates a unidirectional specification toward trophectoderm; however, further cultivation did not lead to the formation of substantial numbers of EVTs, which was the aim. Flow cytometry for HLA-G, an important marker of EVTs, at four time points (hiPSC (D0), differentiation day 2 (D2), D7, and D9), detected low proportions of HLA-G^pos^ cells, without any increase over time, indicating the inefficiency of the published BAP protocol for giving rise to EVTs. Immunofluorescence (IF) imaging at the same time points revealed rare patches of HLA-G^pos^ cells in D9 cultures only (Figure 1D).

We hypothesized that either the inhibition of the activin/NODAL/TGF-β pathway or signaling via the FGF/VEGF receptor in the BAP protocol (Figure 2, Condition A) may not be sufficient to generate EVTs [13]. Moreover, as previously demonstrated for the modulation of cardiac differentiation [24], both cell-to-cell interactions and paracrine factors within the culture may affect differentiation into trophoblast subtypes. Modifications of the above parameters were therefore applied to promote a more efficient generation of HLA-G^pos^ EVTs within our trophoblast differentiation cultures. The applied modifications and their corresponding flow cytometric analysis for HLA-G^pos^ cells after differentiation in a 6-well format are summarized in Figure 2.

Similar to Okae et al. [28], we increased the concentration of the activin/NODAL/TGF-β pathway inhibitor A83-01 to 7.5 µM on day 1 (Condition B). A83-01 at an increased concentration was added on D1, as D0 cells did not tolerate the higher concentration. Although IF imaging on differentiating cultures in a 12-well format indicated distinct HLA-G^pos^ clusters, flow cytometric analysis in parallel 6-well cultures showed comparable numbers of HLA-G^pos^ cells as that with the original BAP protocol (<4%).

Next, we modified seeding densities. In fact, reduced seeding densities (Condition C) resulted in more HLA-G^pos^ cells in both IF and flow cytometric analysis (~11% ± 4.5%, mean, SD). Interestingly, when comparing parallelly treated cells in both formats, with a similar cell density but different bulk density (BCD; cells/mL), via IF imaging, there was a clear difference in both 12-well and 6-well formats, whereby 12-wells had greater proportions of HLA-G^pos^ cells than the 6-well format.

To create similar conditions in both 12-well and 6-well culture formats in terms of BCD, the media volumes were adjusted (Condition D). Equivalent volumes of the medium with respect to the cell numbers led to an increase in the proportion of HLA-G^pos^ cells in flow cytometric analysis from 6-wells (22.8% ± 4.2%, mean, SD).

Lastly, we hypothesized that since EVT cells are post-mitotic, increasing the concentration of PD173074, a potent FGFR1 inhibitor, could inhibit proliferation and may guide cells into a post-mitotic EVT-like state. We found that an increase in PD173074 to 0.4 µM (Condition E) resulted in up to 71.6% HLA-G^pos^ cells. Condition E, hereinafter referred to as “BhAhP” (BMP4, high A83-01, high PD173074), was found to be optimal for generating HLA-G^pos^ cells. The new BhAhP protocol was also applied to two other hiPSC lines, with significant increases in the proportion of HLA-G^pos^ cells (*p* value < 0.05). However, cell line-specific adaptation of seeding densities (cells/surface area) and BCD was required for optimized differentiation. Individual seeding densities and BCDs for each hiPSC line were established (condition E.O and E.M); these sets of optimization experiments resulted in the generation of up to 38.7% and 33.3% HLA-G^pos^ cells, respectively (Appendix A).

### 3.2. Comparative Characterization of iPSC-Derived Trophoblast-Like Cultures after BAP and BhAhP Differentiation

We characterized both BAP- and BhAhP-differentiated cultures on D9 against Lee et al.’s [22] bona fide first-trimester mononuclear trophoblast criteria, with a particular focus on HLA expression. According to Lee et al., CTBs and STBs should be HLA-null, with EVTs expressing surface HLA-G. To investigate surface levels of all HLA class I molecules, we analyzed the cultures via flow cytometry. A pan HLA class I antibody (clone W6/32) was used to detect all HLA class I molecules. Additionally, we also individually investigated HLA -A, -B, -C, -E, and -G surface proteins. Furthermore, individual analysis of HLA-A allele groups was performed with specific antibodies for each hiPSC line. Detection of HLA-B was performed with a mix of antibodies that detect individual allele groups and HLA-Bw4 and Bw6 antibodies, which can, together, detect all HLA-B allele groups (overview of antibodies listed in Appendix A, Appendix A for HLA allele-specific antibodies). Surface HLA-C, -E, and -G were measured with pan antibodies that bind against all allele groups of the respective HLA gene.

Undifferentiated cells (D0), of all hiPSC lines included, were positive for W6/32 (pan HLA class I) and at least one of two HLA-A alleles, while they were negative for HLA-B, -C, -E, and -G (Figure 3A and Appendix A). BAP treatment resulted in a significant reduction in the proportion of cells positive for W6/32 and all individual HLA-A alleles. The modified BhAhP treatment resulted in a smaller reduction in cells expressing any HLA-A allele compared to BAP-treated cells, with the exception of HLA-A*24 in Moo-2 cells.

In fact, the expression of HLA-A alleles varied in each hiPSC line when treated with BhAhP. The proportion of Isis cells expressing either HLA-A alleles (02 and 24) did not show significant changes compared to undifferentiated cells (Figure 3A). A significant reduction in the proportion of both HLA-A alleles (23 and 30) was observed in the Osiris cells during BhAhP treatment. For the Moo-2 cell line, BhAhP treatment had opposing results on the expression of HLA-A alleles, whereby the proportion of cells expressing HLA-A*01, which was surprisingly undetectable in undifferentiated cells, was significantly increased, while HLA-A*24 was significantly reduced and was almost undetectable (Appendix A). In all iPSC lines, no significant increase in any HLA-B allele could be detected after BAP or BhAhP treatment (Figure 3A and Appendix A). Appearance of HLA-C-expressing cells could be only observed at relatively low levels after BhAhP treatment (Figure 3A and Appendix A). Finally, for HLA-G, there was no increase in HLA-G-expressing cells after BAP treatment, with all three cell lines remaining negative after differentiation. In striking contrast, there was a significant increase in HLA-G-expressing cells in all differentiated hiPSC lines after BhAhP treatment (Figure 3A and Appendix A).

KRT7, GATA3, and TFAP2C were also analyzed as protein markers of mononuclear trophoblasts. Bona fide first-trimester mononuclear trophoblast cells should be triple-positive for all three markers. IF imaging of BAP-treated cells indicated the presence of triple-positive regions of KRT7, GATA3, and TFAP2C (Figure 3B). Some very rare HLA-G^pos^ cells could be detected; however, most areas were negative for HLA-G^pos^ cells (Appendix A). BhAhP treatment resulted in many large patches of triple-positive KRT7, GATA3, and TFAP2C cells. In striking contrast to BAP-treated cells, HLA-G^pos^ cells were detectable throughout the BhAhP culture, confirming with the data via flow cytometry (Figure 3A and Appendix A). These HLA-G patches were KRT7^pos^ and GATA3^pos^, but TFAP2C^neg^ (Figure 3C). The presence of human chorionic gonadotrophin β (hCGβ) was confirmed through a lateral flow immunoassay with a 2 day-conditioned medium (Appendix A).

The methylation of the *ELF5* promoter was additionally analyzed. In first-trimester trophoblast cells, this promoter is known to be hypo-methylated, while it is hyper-methylated in undifferentiated hPSCs. The tested hiPSC lines had an average methylation of 75% at 10 CpG sites located upstream of the *ELF5* gene (Figure 3D). BAP treatment of hiPSCs resulted in an efficient demethylation of most CpG sites. Demethylation patterns were uniform amongst all three cell lines, and by D9, BAP cells had an average methylation of 29% ± 3.7% (mean, SD). BhAhP treatment resulted in even more efficient demethylation. BhAhP treatment on the Isis iPSC line resulted in an average methylation of 13.33% ± 3.78% (mean, SD), which was lower than the average methylation of the choriocarcinoma cell line, Jeg3 (19% ± 3.4%), but still slightly higher than JAR (6.75% ± 2.5%) (Figure 3D). Out of the 10 analyzed CpG positions, CpGs at positions −004 and −218 showed the highest methylation rates in comparison to the choriocarcinoma cell lines (Figure 3E). hiPSC lines Moo-2 and Osiris after BhAhP treatment also resulted in a significant demethylation in the *ELF5* promoter region, with overall methylation rates of 22% ± 5.3% and 23% ± 8.66%, respectively (Appendix A).

According to Lee et al. [22], the high expression of four C19MC miRNAs (*miR-517a-3p, miRNA-517b-5p*, *miR-525-3p*, and *miR-526-3p*) relative to *miR-103a* is another characteristic of bona fide first-trimester mononuclear trophoblast cells. Both BAP and BhAhP treatment, however, resulted in the down-regulation of C19MC miRNAs across all three hiPSC lines. In comparison to the choriocarcinoma cell line Jeg3, hiPSCs and BAP/BhAhP-differentiated iPSCs expressed miRNAs 100–10,000-fold lower. *miR-525-3p* and *miR-526-3p* expression became undetectable in all BAP-treated cells, while only *miR-526-3p* was detectable in all BhAhP-treated cells (Figure 3F and Appendix A).

### 3.3. Transcriptome Profiling of iPSC-Derived Trophoblast-Like Cultures after BAP and BhAhP Differentiation

Bulk RNA sequencing (RNA-seq) from the Isis hiPSC line was performed from undifferentiated cells and D9 of differentiation (BAP and BhAhP). There was a total of 4899 differentially expressed genes (DEGs) between D9 BAP trophoblast-like cell cultures and undifferentiated hiPSCs, and 4876 DEGs between D9 BhAhP trophoblast-like cell cultures and hiPSCs.

Overall, BhAhP treatment did result in the up-regulation of more genes known to be “elevated” in trophoblast cells according to the “Human Protein Atlas” (HPA) single-cell RNA sequencing transcriptome (v21.1.proteinatlas.org) [29] than that after BAP treatment. Accordingly, there was less down-regulation of such genes after BhAhP treatment than after BAP treatment. BAP treatment up-regulated 2265 genes, where 416 of those genes overlapped with genes that pass the criteria of “elevated” in trophoblast cells. Of 2510 genes up-regulated in D9 BhAhP-treated cells, 574 overlapped with genes of this category. On the other hand, BAP differentiation down-regulated 2634 genes, with 299 genes representing “elevated trophoblast genes”. BhAhP differentiation down-regulated 2366 genes; among them are 196 “elevated trophoblast genes”. However, 63 of such down-regulated “elevated trophoblast genes” were still highly expressed (normalized count > 8000) in BAP-differentiated cells, while in BhAhP-differentiated samples, 90 down-regulated trophoblast genes were still highly expressed (Figure 4A).

To further investigate BAP and BhAhP treatment in regard to trophoblast subtypes, genes known to be elevated in first-trimester CTBs, STBs, and EVTs according to HPA, were analyzed for differential expression. BAP treatment resulted in the significant up-regulation of 28% CTB, 38% STB, and 26% EVT elevated genes (adjusted *p*-value < 0.01), while significant down-regulating occurred for 24% CTB, 15% STB, and 31% EVT elevated genes compared to the hiPSC state. Substantially, a better match was obtained for D9 differentiated cells after BhAhP treatment. Those cultures showed a significant up-regulation of 36% CTB, 51% STB, and 41% EVT elevated genes. BhAhP treatment also significantly down-regulated fewer trophoblast elevated genes, at only 17% CTB, 8% STB, and 33% EVT elevated genes (Figure 4B).

A direct comparison of the two treatment protocols indicates that BhAhP not only up-regulated a larger number of trophoblast genes, but also led to a higher expression of such genes. While BAP treatment significantly up-regulated 32 out of 45 STB HPA-enriched genes, BhAhP treatment resulted in a significant up-regulation for 35 out of 45. Furthermore, 24 out of 45 STB HPA-enriched genes were significantly higher in BhAhP than BAP treatment (Figure 5A; indicated with *).

When comparing EVT HPA-enriched genes, BhAhP treatment led to a significant up-regulation for 10 out of 26, while BAP treatment only significantly up-regulated 8 out of 26. A direct comparison between both treatments indicated that BhAhP significantly up-regulated 6 out of 26 EVT HPA-enriched genes (Figure 5B; indicated with *). Of particular note is that HLA-G with BAP treatment resulted in a significant −2.3-fold change (log2), while BhAhP resulted in a significant 6.69-fold change (log2).

### 3.4. Characterization of BhAhP-Generated HLA-G^pos^ EVT-Like Trophoblasts after Prolonged Culture

Unlike D9 BAP cultures, D9 BhAhP cells remained adherent. This enabled us to extend the cultures to D21 in order to investigate whether the culture can be maintained under these conditions or may even develop into more mature trophoblasts. We performed IF imaging on Isis D21 cultures and found that cultures are still HLA-G^pos^, and that the regions maintained co-localization with KRT7 and GATA3 (Figure 6A). Flow cytometric analysis on D21 revealed a significant increase in HLA-G protein expression in the HLA-G^pos^ cell population compared to that on D9 (Figure 6B). Analysis of bulk RNA expression showed significant changes in gene expression in only 8 out of 26 EVT-enriched genes in Isis-derived trophoblast cells (Figure 6C; indicated with *). Though, most importantly, a prolonged culture did not result in significant changes in bulk HLA-G expression. This suggested that prolonged BhAhP treatment allowed for the maintenance of the expression characteristics to a wide extent.

Next, we analyzed the presence of other surface HLA class I molecules in Isis-derived BhAhP-treated HLA G^pos^ cells on D9 and D21. Flow cytometry indicated that at both time points, all HLA-G^pos^ cells co-stained with W6/32. On D9, the majority of HLA G^pos^ cells also co-stained with either HLA-A allele, while a very minimal proportion of cells were double-positive with HLA -B, -C, or -E. The effects of prolonged exposure on D21 Isis BhAhP-derived HLA-G^pos^ cells meant that there was a significant reduction in one HLA-A allele compared to D9 cultures (HLA-A24, 95% ± 1% to 53% ± 36%), while HLA -B, -C, and -E remained absent at both time points (Figure 6D).

### 3.5. D21 BhAhP Cultures Contain Binucleated HLA-G^pos^ Cells That May Indicate the Formation of Trophoblast Giant-Like Cells

Microscopic observations of D21 BhAhP cultures revealed rare patches of large cells. Many of the cells within these patches were HLA-G^pos^, and further staining with a nuclear dye (DAPI) and a membrane protein (ECAD) antibody clearly demonstrated that these cells were binucleated (Figure 7A). The only terminally differentiated multinucleated HLA-G^pos^ cells in the placenta that could be so large are the placental bed giant cells. Using the “Descartes” single-cell RNA-seq transcriptome from the Brotman Baty Institute (BBI) [30], we investigated the top 100 genes expressed in their “Trophoblast Giant” cluster and compared the fold-changes of these genes in D9 BhAhP–hiPSC and D21 BhAhP–hiPSC samples. D9 BhAhP significantly down-regulated 5 genes, while significantly up-regulating 65 genes; D21 BhAhP significantly down-regulated 3 genes and significantly up-regulated 67 genes. Comparing the top 5 most abundantly expressed genes in the BBI “Trophoblast Giant” cluster (*CSH1* (84,000 TPM), *TFPI2* (23,000 TPM), *CSH2* (22,000 TPM), *PAPPA* (20,000 TPM), and *ADAM12* (15,000 TPM)), D21 BhAhP significantly up-regulates all of these genes, while D9 only significantly up-regulates *TFPI2*, *PAPPA*, and *ADAM12*.

## 4. Discussion

Thus far, the published BAP protocols for hiPSC-derived trophoblast cells yield limited numbers of HLA-G^pos^ cells, and the resultant cultures resemble the STB subtype the most. This limits their use in terms of elucidating the underlying mechanisms involving EVT and CTB, like immunomodulation during pregnancy or differentiation into trophoblast subtypes, respectively. We hypothesized that further optimization of the BAP differentiation protocol would give rise to EVT cells, which we defined based on the presence of surface HLA-G, in contrast to CTB and STB being HLA-null. HLA-G expression on EVTs is of high functional relevance because of its direct inhibitory effect on T, B, NK, and antigen-presenting cells by interacting with several leukocyte receptors such as KIR2DL4, ILT2, ILT4, and CD8 [31]. Even though HLA-G by itself does not account for all the immunomodulation properties of EVTs, as demonstrated by Tilburgs et al. [19], in vitro experiments with HLA-G transfects have demonstrated its immunomodulation effects [32,33,34,35]. This is further supported by the observation that aberrant HLA-G presence after organ transplantation is associated with a greater chance of graft acceptance [36]. This reinforces that EVTs are highly specialized cells.

In general, recent publications either use the generation of trophoblast stem cells (TSC) as an intermediate during differentiation [37,38,39,40] or the two-step differentiation protocol with a conditioned medium [41]. We avoided the TSC step and opted to develop a direct differentiation protocol, as well as only implementing fully defined media during the entire differentiation process, with E8 for the pluripotent phase and E6 for differentiation. The simplicity and the fully defined manner of the medium make it ideal for protocol optimization, especially since mouse embryonic fibroblast-conditioned media (MEF-CM) is disadvantaged by inconsistent concentrations and molecules [42].

During the stepwise optimization, we considered that invasive EVT cells migrate out of the cell column, which results in less contact to neighboring trophoblast cells and therefore less paracrine effects, and that EVTs exist in a lower oxygen tension environment. Therefore, to potentially create a more physiological environment for EVT cells, we reduced the seeding density, as well as increased the volume of the medium. These steps favored the generation of HLA-G^pos^ cells during BhAhP differentiation.

Once optimal cell seeding densities had been determined for each hiPSC line, the differentiation medium always contained the same three components throughout the process, of which the concentration was adjusted. In contrast to many reports for generating HLA-G^pos^ cells using the EVT-differentiation medium from Okae et al. [28,43,44,45], we omitted Neuregulin 1 (NRG1), a molecule promoting EVT formation in placental explant cultures [46]. Our finding that NRG1 may not be necessary for the differentiation of primed hPSCs into EVTs is consistent with the HLA-G^pos^ protocol from Horii et al. [41], which also omits NRG1. However, their two-step differentiation protocol requires MEF-CM, which introduces batch-to-batch variability and only resulted in a maximum of 25% HLA-G^pos^ cells. This emphasizes the advantage of our fully defined BhAhP protocol generating significantly higher HLA-G proportions of up to 71.6%.

When comparing hiPSC-derived cells from BAP versus BhAhP treatment against the most rigorous criteria for first-trimester trophoblasts [22], we found a lower ELF5 promoter methylation in the BhAhP protocol and co-expression of HLA-G with KRT7 or GATA3 only after BhAhP treatment, which underlines the advantages of our protocol.

Similar to EVTs derived from primary trophoblast stem cells, which express both surface HLA-A and -B [47], and to other trophoblast cultures derived from primed hPSCs, our BhAhP-derived HLA-G^pos^ cells do not completely fulfill the criteria for trophoblasts as proposed by Lee et al. [22]. Human primed PSC-derived trophoblast-like cells, however, have shown to recapitulate early pregnancy events, indicating that only certain characteristics of trophoblast cells would be sufficient depending on the asked biological question [48,49].

A recent paper demonstrates the importance of C19MC miRNAs. Activation of dormant C19MCs in both primed hESC and primed hESC-derived human trophoblast stem-like cells rescued proliferation and differentiation capacity [50]. Interestingly, we observed that our primed hiPSC-derived trophoblast-like cultures do not express members of this miRNA cluster. While our focus was not on trophoblast stem cells, we could demonstrate that our primed iPSC-derived trophoblast cultures contained different trophoblast subtypes that very closely resembled primary trophoblast cells. Our differentiation protocol may just bypass the trophoblast stem cells intermediate state where these miRNAs may be required. This is supported by the first-trimester trophoblast cell line HTR-8, which also does not express C19MC miRNAs [51].

Another important aspect that was not addressed so far is the verification of other HLA class I molecules found on hiPSC-derived HLA-G^pos^ cells. Although HLA-A expression appears to be decreased during extended culture in Isis-derived trophoblast cultures, our BhAhP-derived HLA-G^pos^ cells still expressed well-detectable levels of HLA-A on D21. In addition, HLA-C, another EVT marker that plays an important role in EVT tolerance and immunity [52,53], was just slightly expressed in D9 samples and also did not increase on D21. The cells generated here did not display the full range of ligands that EVT cells use to interact with the immune system, as both HLA-C and HLA-E were not present on the surface of our HLA-G^pos^ cells. Whether remaining levels of HLA-A expression and the lack of HLA-C in HLA-G^pos^ cells, on both BAP and BhAhP trophoblast cultures may be connected to the lack of expression of the C19MC cluster remains to be explored. This would need further investigation and potential adaptation of the differentiation protocol.

Although not entirely matching the most rigorous criteria for first-trimester trophoblasts [22], our BhAhP-derived cultures express increased levels of various gene characteristic for trophoblasts, even more than BAP-treated cells. The transcriptome of BAP-treated hiPSCs has already been reported to be clearly that of trophoblasts [54], with a bias toward STB [15]. From our analysis, BhAhP treatment up-regulates even more gene characteristics for every trophoblast subtype than BAP treatment. Furthermore, BhAhP treatment results in a higher up-regulation of STB and EVT “cell-type enriched” genes. This potentially indicates that the original concentrations of BAP treatment pass the activation threshold of trophoblast genes, while the higher concentrations in our protocol accelerate and activate more trophoblast genes. This effect is most clearly demonstrated in the STB genes, with more than half being significantly higher-expressed after BhAhP treatment.

Interestingly, we could also observe rare patches of multinucleated HLA-G^pos^ cells, which is indicative of the fusion or acytokinetic cell division of EVT cells. This process is known to lead to the formation of multinucleated trophoblast giant cells (MTGCs). While interstitial EVTs are also multinucleated, the observed cells here are large and polygonal, while interstitial EVTs are small spindle-shaped cells. MTGCs contain two or more nuclei with varying sizes, have an extensive cytoplasm with a diameter of 50–100 µm, and are mostly found deep in the myometrium at the uterine placental implantation site [55]. To our knowledge, this is the first report on hPSC-derived HLA-G^pos^ multinucleated trophoblast cells. Given that our cells are ECAD^pos^, we assume they are still in the early stages of placental development, as ECAD^pos^ distal EVTs can be found at week 10, but become ECAD-negative by week 22 [56]. MTGC in normal pregnancy usually demarcates the furthest point of EVT invasion. Defects in EVT invasion lead to abnormal placentation and therefore pregnancy complications. These cells may allow for the investigation of pathomechanisms of placental complications related to MTGCs, e.g., preeclampsia [57].

Even though there have been many recent reports indicating the superiority of naïve hPSC in generating trophoblast cells and especially after transitioning into trophoblast stem cells, primed hPSCs are more abundant, well established, and widely used among researchers. Furthermore, it is much cheaper to culture primed hPSCs versus naïve/trophoblast stem cells. Moreover, the trophoblast stem cell medium is not available commercially and hence limits accessibility.

In conclusion, the directed differentiation of primed hiPSCs can circumvent the difficulty of acquiring biologically relevant numbers of HLA-G^pos^ EVTs to elucidate disease mechanisms and immunomodulatory properties. While BhAhP cells do not entirely match the criteria for the definition of trophoblasts, as recently proposed by Lee et al. [22], these cells will allow for an on-demand access of directly differentiated HLA-G^pos^ cells with trophoblast-like characteristics. This differentiation protocol can be easily scaled out, circumventing the difficulties in obtaining primary human HLA-G^pos^ EVTs, or without the long and complex process of generating human trophoblast stem cells [58].

## Figures and Tables

**Figure 1 cells-12-02070-f001:**
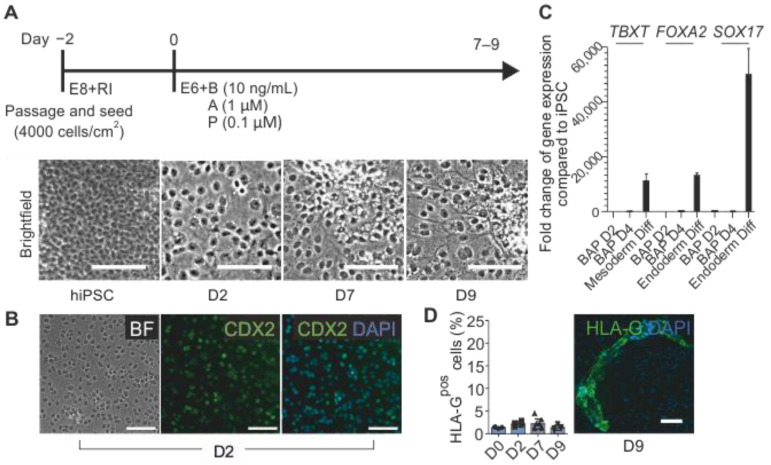
Schematic of the BAP differentiation protocol for the derivation of trophoblast-like cells from hiPSCs. (**A**) hiPSCs are passaged two days before the initiation of BAP treatment. BAP treatment results in the cells adopting a flattened trophoblast-like morphology by D2. E8, Essential 8™; RI, Rho kinase inhibitor; E6, Essential 6™; B, BMP4; A, A83-01; P, PD173074. (**B**) IF image of CDX2 presence on D2. Scale bars represent 100 µm. (**C**) Bar chart indicating the fold-changes of the early mesoderm genes, *TBXT*, and early endoderm genes, *FOXA2* and *SOX17*, upon differentiation compared to hiPSCs. (**D**) HLA-G^pos^ cells are rare in BAP-treated cultures and can only be found via IF imaging at the end of differentiation. Data represent the mean ± SD. Scale bars represent 100 µm.

**Figure 2 cells-12-02070-f002:**
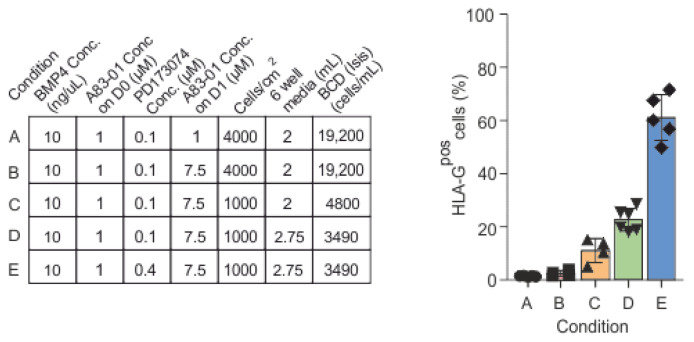
Optimization of the published BAP protocol leads to an improved differentiation of hiPSCs into HLA-G^pos^ trophoblast-like cells. Optimization conditions for different parameters to the original BAP protocol to increase the proportion of HLA-G^pos^ cells in the Isis cell line. Bar chart representing the increase in HLA-G^pos^ cells with each optimization step. Data represent the mean ± SD.

**Figure 3 cells-12-02070-f003:**
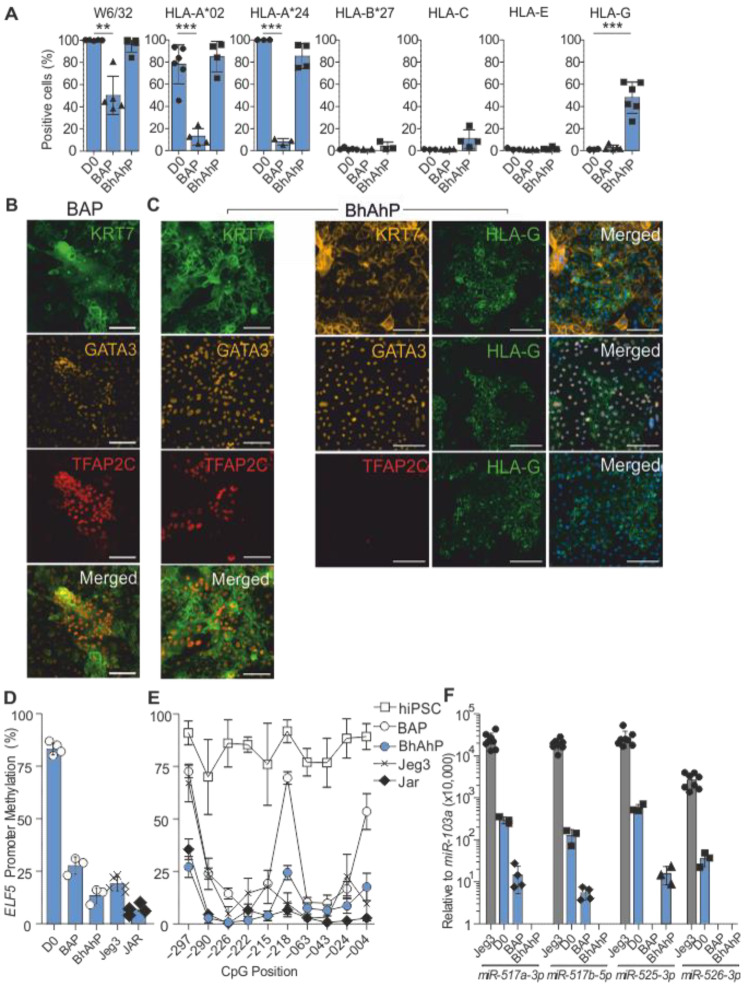
D9 BAP- and BhAhP-differentiated cells express certain characteristics of first-trimester trophoblasts. (**A**) Bar charts indicating the percentage of HLA^pos^ cells for hiPSC (D0), D9 BAP, and D9 BhAhP-treated samples from the Isis iPSC line. Data represent the mean ± SD; ** *p* < 0.01, *** *p* < 0.0001 (*t*-test). (**B**) IF imaging of D9 BAP-differentiated hiPSCs indicates triple-positive regions KRT7, GATA3, and TFAP2C. Scale bars represent 100 µm. (**C**) IF imaging of D9 BhAhP-treated hiPSCs indicates triple-positive regions KRT7, GATA3, and TFAP2C, and that regions of HLA-G^pos^ cells co-localize with KRT7 and GATA3, but not with TFAP2C. Scale bars represent 100 µm. (**D**) Bar chart of average methylation across 10 CpG positions upstream of *ELF5* on D0, D9 BAP- and D9 BhAhP-treated Isis hiPSCs, Jeg3, and JAR cells. Data represent the mean ± SD. (**E**) Bisulfite PCR analysis on the *ELF5* promoter in Isis hiPSCs, D9 BAP- and D9 BhAhP-treated Isis hiPSC, and the choriocarcinoma cell lines, Jeg3 and JAR. Data represent the mean ± SD. (**F**) Normalized expression of four miRNAs from C19MC in the primary choriocarcinoma cell line, Jeg3, Isis hiPSC, and D9 BAP- and D9 BhAhP-treated Isis hiPSC. Results are normalized to *miR-103a* and multiplied 10,000× to ensure all logged values are positive. Data represent the mean ± SD.

**Figure 4 cells-12-02070-f004:**
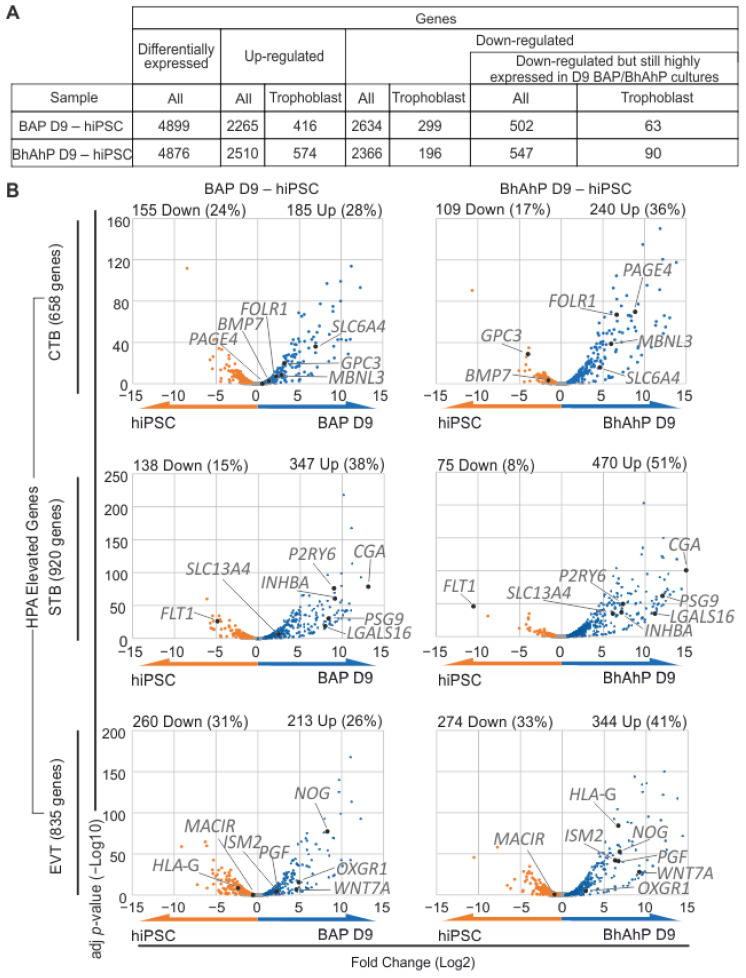
The transcriptome of BhAhP-treated iPSC derivatives is more similar to CTBs, STBs, and EVTs from 6–14 week placenta than BAP-treated cells. (**A**) Overview of the analyzed RNA-seq results for D9 BAP–hiPSC and D9 BhAhP–hiPSC samples. The table includes the number of genes that were differentially expressed compared to undifferentiated iPSCs, up- and down-regulated, and the number of down-regulated genes that are still highly expressed (normalized count > 8000). Genes “elevated” in trophoblast cells in accordance with The Human Protein Atlas (HPA) single-cell RNA-seq transcriptome criteria from placentas between weeks 6 and 14 were considered trophoblast genes. (**B**) Volcano plots of genes “elevated” in CTB, STB, and EVT cells (HPA) in D9 BAP–hiPSC and D9 BhAhP–hiPSC samples. Grey dots indicate non-significant fold-change (adjusted *p*-value > 0.05).

**Figure 5 cells-12-02070-f005:**
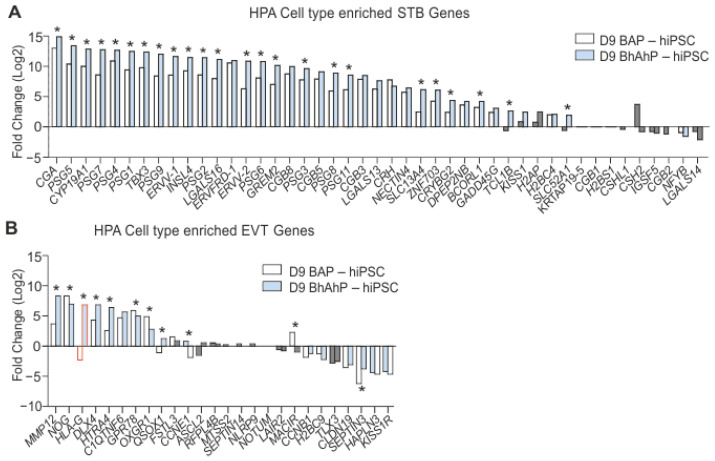
BhAhP treatment increases the expression of STB and EVT genes. (**A**) Bar chart of the fold-change (log2) of genes enriched in STB (HPA). (**B**) Bar chart of the fold-change (log2) of genes enriched in EVT (HPA). Grey bars indicate non-significant fold-change between BAP/BhAhP-treated cultures and hiPSCs (adjusted *p*-value > 0.05); * indicates significant fold-change (adjusted *p*-value < 0.05) between the D9 BAP- and D9 BhAhP-treated samples.

**Figure 6 cells-12-02070-f006:**
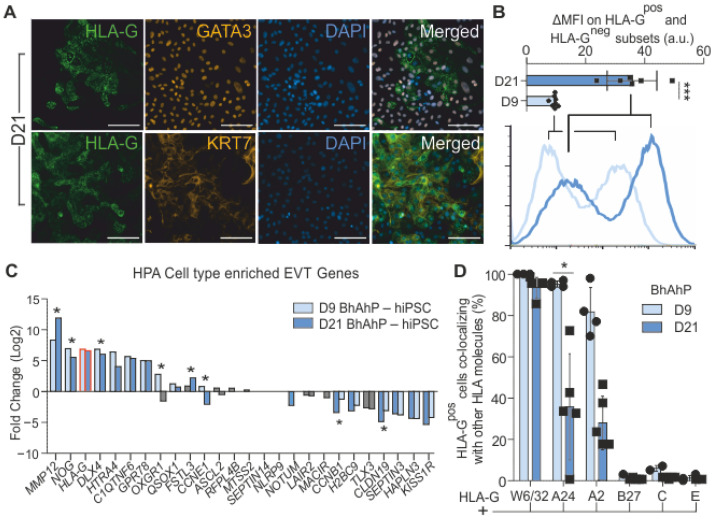
D21 BhAhP cultures still have HLA-G surface expression. (**A**) IF images of D21 BhAhP-treated Isis-derived trophoblast cultures indicate that regions of HLA-G^pos^ cells are still present and still co-localize with GATA3 and KRT7. Scale bars represent 100 µm. (**B**) Bar chart and a representative histogram of the delta median fluorescent intensity of HLA-G-positive and -negative subsets (ΔMFI^pos-neg^) of D9 and D21 BhAhP-treated cells. Prolonged treatment results in an increased surface HLA-G. a.u.—Arbitrary units. Data represent mean ± SD; *** *p* < 0.0001 (*t*-test). (**C**) Bar chart of the fold-change (log2) of genes enriched in EVT (HPA) in D21/D9 BhAhP-treated and hiPSC Isis cells. Asterisks indicate significant fold-change (adjusted *p*-value < 0.05) between the D9 and D21 BhAhP-treated samples. Grey bars indicate non-significant fold-change between D9/D21 BhAhP (adjusted *p*-value > 0.05). (**D**) Bar chart indicating the percentage of HLA-G^pos^ cells that are also positive for other HLA class I molecules. Data represent the mean ± SD; * *p* < 0.05 (*t*-test).

**Figure 7 cells-12-02070-f007:**
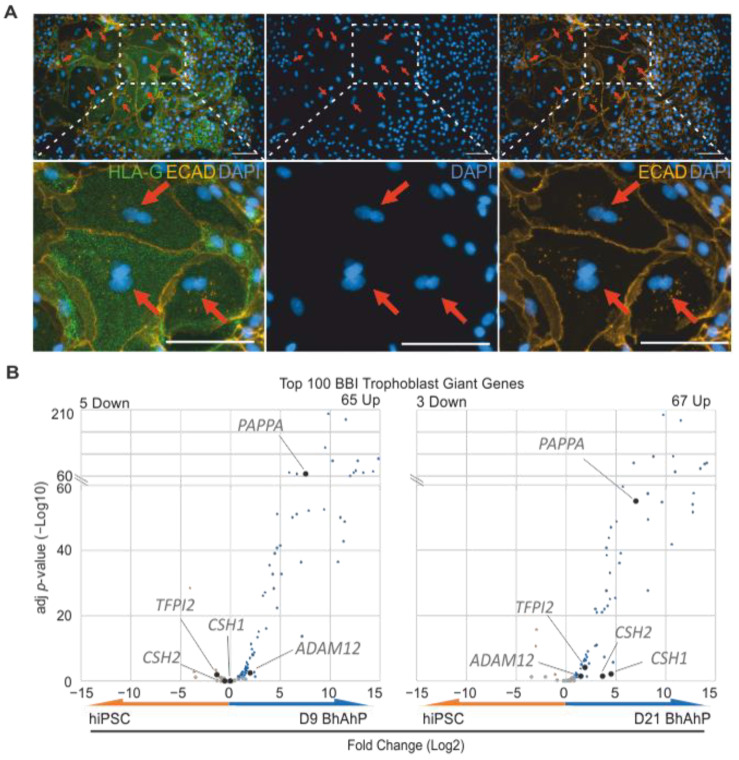
Rare appearance of multinucleated HLA-G^pos^ cells after extended BhAhP treatment. (**A**) IF images of HLA-G^pos^ and ECAD^pos^ D21 BhAhP-treated hiPSCs. Second row are enlarged regions of interest. Multinucleated cells are indicated with red arrows. Scale bars represent 100 µm. (**B**) Volcano plots of the top 100 genes expressed in “Trophoblast Giant” from the Descartes single-cell RNA-seq transcriptome (BBI) in D9 BhAhP–hiPSC and D21 BhAhP–hiPSC samples. Grey dots indicate non-significant fold-changes compared to undifferentiated hiPSCs (adjusted *p*-value > 0.05).

## Data Availability

All sequencing data have been deposited in the NCBI Gene Expression Omnibus (GEO) public repository under accession number GSE234949.

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
