# Peer review of "An Improved Protocol for Targeted Differentiation of Primed Human Induced Pluripotent Stem Cells into HLA-G-Expressing Trophoblasts to Enable the Modeling of Placenta-Related Disorders"

_cells, 2023, doi:10.3390/cells12162070_

Round 1

Reviewer 1 Report

The manuscript “An improved protocol for targeted differentiation of human iP-SCs into HLA-G expressing trophoblasts cells enabling the modelling of placenta-related disorders” by Shum et al. tries to improve the targeted differentiation of iPSC cells towards the trophoblastic lineages. The concept of this study is highly interesting, to use iPSC cells to model placenta-related disorders, because most of the pregnancy related disorders like preeclampsia or HELLP-syndrome have no appropriate cell culture nor animal model. The study found an improved differentiation protocol to generate HLA-Gpos EVTs by modifying slightly the differentiation cocktail. The rationale of this research article is of interest, but the data remain descriptive and lack the translation into a functional implementation as suggested by the title. Additionally, the manuscript lacks state of the art methods, like establishing a placenta organoid model with their iPSCs cells as shown for several cell lines in the recent years and the manuscript lacks a discussion with the most recent literature in this field.

Major concerns:

-       - A general remark on the manuscript exemplified by line 522: All cited manuscripts concerning the differentiation of iPSCs into trophoblastic cells are published 2013-2016. There are multiple studies published in the years 2021-2023 using highly modified differentiation protocols to model for example preeclampsia. The authors should integrate the newest literature and discuss the benefits and disadvantages of their novel protocol.

-       -The study lacks any evidence for fully functional differentiated cells, e.g., β-HCG secretion (STB), Tube formation, MMP secretion, migration/Invasion behavior (EVTs) etc.

-       -The main finding that the condition E led to an increased differentiation to a total of 71.6% is perfect in line with the actual literature showing differentiation efficiencies of 60.9% up to 78.2% with iPSC cells, which can be increased by using embryonic stem cells up to 95.0%.

-       -In Figure 1D the differentiation percentage of the standard BAP protocol is surprisingly low compared with the literature, suggesting a problem with the initial protocol, culture conditions, handling of the cells or the cells viability.

Author Response

Reviewer comment: The manuscript “An improved protocol for targeted differentiation of human iPSCs into HLA-G expressing trophoblasts cells enabling the modelling of placenta-related disorders” by Shum et al. tries to improve the targeted differentiation of iPSC cells towards the trophoblastic lineages. The concept of this study is highly interesting, to use iPSC cells to model placenta-related disorders, because most of the pregnancy related disorders like preeclampsia or HELLP-syndrome have no appropriate cell culture nor animal model. The study found an improved differentiation protocol to generate HLA-Gpos EVTs by modifying slightly the differentiation cocktail. The rationale of this research article is of interest, but the data remain descriptive and lack the translation into a functional implementation as suggested by the title. Additionally, the manuscript lacks state of the art methods, like establishing a placenta organoid model with their iPSCs cells as shown for several cell lines in the recent years and the manuscript lacks a discussion with the most recent literature in this field.

The aim of our study was to establish an iPSC-based protocol for the generation of HLA-G expressing trophoblast cells from primed iPSCs, which are well established and widely used by researchers. We now edited the title to reflect this. The strength of our protocol is the simplicity regarding medium composition and handling. The applicability of primed iPSC and the bypass of the trophoblast stem cells (TSC) intermediate step makes this  protocol broadly applicable for researchers. Certainly, placenta organoid models are of high relevance, but their application is dependent on the scientific context you want to address. Organoids are not always a suitable cell model system, and lacks the defined nature of 2D cultures. In addition, placenta organoids require first trimester placentas (Turco et al. 2018, Sheridan et al. 2020, Haider et al. 2018), which are not applicable for everybody. Alternatively, trophoblast organoids from naïve stem cells have been reported (Karvas et al., 2022) but the lengthy and advance process of naïve stem cell conversion reduces the availability. This point is also further explained below. For discussion of the most recent literature please read our comments below.

Major concerns:

Reviewer comment: A general remark on the manuscript exemplified by line 522: All cited manuscripts concerning the differentiation of iPSCs into trophoblastic cells are published 2013-2016. There are multiple studies published in the years 2021-2023 using highly modified differentiation protocols to model for example preeclampsia. The authors should integrate the newest literature and discuss the benefits and disadvantages of their novel protocol.

The reviewer is correct; at this specific site, we are just citing the older manuscripts. However, the statement holds also true for the newer publications, which we now added to the manuscript. We wanted to emphasize the advantage of our protocol regarding the bypass of the trophoblast stem cells (TSC) intermediate step and the application of a fully defined medium. Please see our changes in the manuscript from line 524.

In general most, if not all, of the protocols which were published in the years 2021-2023 use the same Okae et al. (2018) EVT protocol with the exception of Soncin et al. (2022), who used the Horii et al. (2019) two step protocol with conditioned medium. Since Okae et al. (2018), the field transitioned towards generating trophoblast stem cells (TSC) as an intermediate during the differentiation of naïve human pluripotent stem cells (PSCs). Castel and David (2022) even showed the differentiation of primed hPSCs with the TSC intermediate step. However, the process requires advanced cell culture technique and it takes 2 months to convert naïve/extended cells into TSC and additional 2-4 weeks to convert primed stem cells into naïve. Furthermore, even after converting into TSC from primed/naïve pluripotent stem cells, the differentiated EVT cells have varying degrees of HLA-Gpos cells. HLA-G quantification after EVT differentiated from converted hPSC to TSC from Io et al. (2021), was 47%, Gao et al. (2019), was 20.2%, Guo et al. (2021), – no quantification, Mischler et al. (2021) was 96.9%, Wei et al. (2022), was 88.8% and 66.7%, Soncin et al. (2022), was 50.3%. In terms of direct differentiation of primed pluripotent stem cells, there are two main protocols. The first from Horii et al., (2016; 2019) which has a reported HLA-Gpos proportion of 14.4% and 25%, respectively and used conditioned medium, which we wanted to avoid. The second is the BAP protocol from Amita et al., (2013), who do not report any quantified HLA-Gpos proportions. In fact, Wei et al. (2017), Lee et al. (2016), Koel et al. (2018), Yabe et al. (2016), Yang et al. (2015), and Sheridan et al. (2019) all use BAP, but do not quantify the proportion of HLA-Gpos cells. Only a review from Roberts et al., (2018) reports a quantified value, citing Yang et al. (2015), at 4.2% when differentiating primed stem cells with BAP. Though, as mentioned in the discussion (from line 524 and line 540) we avoid conditioned medium and can generate a higher maximal HLA-Gpos proportion. The benefits of our protocol without the TSC step is discussed line 608 onwards.

Reviewer comment: The study lacks any evidence for fully functional differentiated cells, e.g., β-HCG secretion (STB), Tube formation, MMP secretion, migration/Invasion behavior (EVTs) etc.

hCG cells were not screened for in the new protocol, as the main objective was to increase the proportion of HLA-Gpos cells in differentiating cultures. However, an immune-lateral flow assay tests positive for at least 25 UI/mL of hCG in 2 day conditioned medium, which would be enough to trigger a pregnancy test. We included this result now in the Supplemental Figure S2.

Reviewer comment: The main finding that the condition E led to an increased differentiation to a total of 71.6% is perfect in line with the actual literature showing differentiation efficiencies of 60.9% up to 78.2% with iPSC cells, which can be increased by using embryonic stem cells up to 95.0%.

      The reviewer is right, there are publications indicating highly efficient differentiations into HLA-Gpos cells, however like mentioned above, these highly efficient protocols apply either naïve pluripotent stem cells or a TSC conversion first, which we do not require with our protocol. The use of primed pluripotent stem cells, which as mentioned are more broadly available, is now reflected in the changed title.

The only publication to report a quantified value of HLA-Gpos proportions after directly differentiating primed human pluripotent stem cells was the review by Roberts et al. (2018), after they heighten pluripotency with BMP4. They reported 77.9% HLA-Gpos cells after BAP differentiation, however they used the antibody clone 4H84, which is known to non-specifically bind other HLA Class I molecules (Polakova et al., 2004). That is why we think that this reported amount might be overestimated. 

Reviewer comment: In Figure 1D the differentiation percentage of the standard BAP protocol is surprisingly low compared with the literature, suggesting a problem with the initial protocol, culture conditions, handling of the cells or the cells viability.

As already stated at the first point: The publications which use the BAP protocol from Amita et al., (2013) are Wei et al. (2017), Lee et al. (2016), Koel et al. (2018), Yabe et al. (2016), Yang et al. (2015), and Sheridan et al. (2019). They all do not quantify the proportion of HLA-Gpos cells. Lee et al. (2016), did perform flow cytometry but did not report the value (Figure 5C in the publication would indicate marginal increase). Only a review from Roberts et al., (2018) reports a quantified value, citing Yang et al. (2015), at 4.2% when differentiating primed stem cells with BAP. This is in line with our findings.

References [1-28]

  1. Amita, M.; Adachi, K.; Alexenko, A.P.; Sinha, S.; Schust, D.J.; Schulz, L.C.; Roberts, R.M.; Ezashi, T. Complete and unidirectional conversion of human embryonic stem cells to trophoblast by BMP4. Proc Natl Acad Sci U S A 2013, 110, E1212-1221, doi:10.1073/pnas.1303094110.
  2. Castel, G.; David, L. Induction of human trophoblast stem cells. Nat Protoc 2022, 17, 2760-2783, doi:10.1038/s41596-022-00744-0.
  3. Castel, G.; Meistermann, D.; Bretin, B.; Firmin, J.; Blin, J.; Loubersac, S.; Bruneau, A.; Chevolleau, S.; Kilens, S.; Chariau, C., et al. Induction of Human Trophoblast Stem Cells from Somatic Cells and Pluripotent Stem Cells. Cell Rep 2020, 33, 108419, doi:10.1016/j.celrep.2020.108419.
  4. Cui, K.; Zhu, Y.; Shi, Y.; Chen, T.; Wang, H.; Guo, Y.; Deng, P.; Liu, H.; Shao, X.; Qin, J. Establishment of Trophoblast-Like Tissue Model from Human Pluripotent Stem Cells in Three-Dimensional Culture System. Adv Sci (Weinh) 2022, 9, e2100031, doi:10.1002/advs.202100031.
  5. Dong, C.; Beltcheva, M.; Gontarz, P.; Zhang, B.; Popli, P.; Fischer, L.A.; Khan, S.A.; Park, K.M.; Yoon, E.J.; Xing, X., et al. Derivation of trophoblast stem cells from naive human pluripotent stem cells. Elife 2020, 9, doi:10.7554/eLife.52504.
  6. Gao, X.; Nowak-Imialek, M.; Chen, X.; Chen, D.; Herrmann, D.; Ruan, D.; Chen, A.C.H.; Eckersley-Maslin, M.A.; Ahmad, S.; Lee, Y.L., et al. Establishment of porcine and human expanded potential stem cells. Nat Cell Biol 2019, 21, 687-699, doi:10.1038/s41556-019-0333-2.
  7. Guo, G.; Stirparo, G.G.; Strawbridge, S.E.; Spindlow, D.; Yang, J.; Clarke, J.; Dattani, A.; Yanagida, A.; Li, M.A.; Myers, S., et al. Human naive epiblast cells possess unrestricted lineage potential. Cell Stem Cell 2021, 28, 1040-1056 e1046, doi:10.1016/j.stem.2021.02.025.
  8. Haider, S.; Meinhardt, G.; Saleh, L.; Kunihs, V.; Gamperl, M.; Kaindl, U.; Ellinger, A.; Burkard, T.R.; Fiala, C.; Pollheimer, J., et al. Self-Renewing Trophoblast Organoids Recapitulate the Developmental Program of the Early Human Placenta. Stem Cell Reports 2018, 11, 537-551, doi:10.1016/j.stemcr.2018.07.004.
  9. Horii, M.; Bui, T.; Touma, O.; Cho, H.Y.; Parast, M.M. An Improved Two-Step Protocol for Trophoblast Differentiation of Human Pluripotent Stem Cells. Curr Protoc Stem Cell Biol 2019, 50, e96, doi:10.1002/cpsc.96.
  10. Horii, M.; Li, Y.; Wakeland, A.K.; Pizzo, D.P.; Nelson, K.K.; Sabatini, K.; Laurent, L.C.; Liu, Y.; Parast, M.M. Human pluripotent stem cells as a model of trophoblast differentiation in both normal development and disease. Proc Natl Acad Sci U S A 2016, 113, E3882-3891, doi:10.1073/pnas.1604747113.
  11. Io, S.; Kabata, M.; Iemura, Y.; Semi, K.; Morone, N.; Minagawa, A.; Wang, B.; Okamoto, I.; Nakamura, T.; Kojima, Y., et al. Capturing human trophoblast development with naive pluripotent stem cells in vitro. Cell Stem Cell 2021, 28, 1023-1039 e1013, doi:10.1016/j.stem.2021.03.013.
  12. Karvas, R.M.; Khan, S.A.; Verma, S.; Yin, Y.; Kulkarni, D.; Dong, C.; Park, K.M.; Chew, B.; Sane, E.; Fischer, L.A., et al. Stem-cell-derived trophoblast organoids model human placental development and susceptibility to emerging pathogens. Cell Stem Cell 2022, 29, 810-825 e818, doi:10.1016/j.stem.2022.04.004.
  13. Koel, M.; Vosa, U.; Krjutskov, K.; Einarsdottir, E.; Kere, J.; Tapanainen, J.; Katayama, S.; Ingerpuu, S.; Jaks, V.; Stenman, U.H., et al. Optimizing bone morphogenic protein 4-mediated human embryonic stem cell differentiation into trophoblast-like cells using fibroblast growth factor 2 and transforming growth factor-beta/activin/nodal signalling inhibition. Reprod Biomed Online 2017, 35, 253-263, doi:10.1016/j.rbmo.2017.06.003.
  14. Lee, C.Q.; Gardner, L.; Turco, M.; Zhao, N.; Murray, M.J.; Coleman, N.; Rossant, J.; Hemberger, M.; Moffett, A. What Is Trophoblast? A Combination of Criteria Define Human First-Trimester Trophoblast. Stem Cell Reports 2016, 6, 257-272, doi:10.1016/j.stemcr.2016.01.006.
  15. Li, Y.; Moretto-Zita, M.; Soncin, F.; Wakeland, A.; Wolfe, L.; Leon-Garcia, S.; Pandian, R.; Pizzo, D.; Cui, L.; Nazor, K., et al. BMP4-directed trophoblast differentiation of human embryonic stem cells is mediated through a DeltaNp63+ cytotrophoblast stem cell state. Development 2013, 140, 3965-3976, doi:10.1242/dev.092155.
  16. Mischler, A.; Karakis, V.; Mahinthakumar, J.; Carberry, C.K.; San Miguel, A.; Rager, J.E.; Fry, R.C.; Rao, B.M. Two distinct trophectoderm lineage stem cells from human pluripotent stem cells. J Biol Chem 2021, 296, 100386, doi:10.1016/j.jbc.2021.100386.
  17. Okae, H.; Toh, H.; Sato, T.; Hiura, H.; Takahashi, S.; Shirane, K.; Kabayama, Y.; Suyama, M.; Sasaki, H.; Arima, T. Derivation of Human Trophoblast Stem Cells. Cell Stem Cell 2018, 22, 50-63 e56, doi:10.1016/j.stem.2017.11.004.
  18. Roberts, R.M.; Ezashi, T.; Sheridan, M.A.; Yang, Y. Specification of trophoblast from embryonic stem cells exposed to BMP4. Biol Reprod 2018, 99, 212-224, doi:10.1093/biolre/ioy070.
  19. Sheridan, M.A.; Fernando, R.C.; Gardner, L.; Hollinshead, M.S.; Burton, G.J.; Moffett, A.; Turco, M.Y. Establishment and differentiation of long-term trophoblast organoid cultures from the human placenta. Nat Protoc 2020, 15, 3441-3463, doi:10.1038/s41596-020-0381-x.
  20. Sheridan, M.A.; Yang, Y.; Jain, A.; Lyons, A.S.; Yang, P.; Brahmasani, S.R.; Dai, A.; Tian, Y.; Ellersieck, M.R.; Tuteja, G., et al. Early onset preeclampsia in a model for human placental trophoblast. Proc Natl Acad Sci U S A 2019, 116, 4336-4345, doi:10.1073/pnas.1816150116.
  21. Soncin, F.; Morey, R.; Bui, T.; Requena, D.F.; Cheung, V.C.; Kallol, S.; Kittle, R.; Jackson, M.G.; Farah, O.; Chousal, J., et al. Derivation of functional trophoblast stem cells from primed human pluripotent stem cells. Stem Cell Reports 2022, 17, 1303-1317, doi:10.1016/j.stemcr.2022.04.013.
  22. Turco, M.Y.; Gardner, L.; Kay, R.G.; Hamilton, R.S.; Prater, M.; Hollinshead, M.S.; McWhinnie, A.; Esposito, L.; Fernando, R.; Skelton, H., et al. Trophoblast organoids as a model for maternal-fetal interactions during human placentation. Nature 2018, 564, 263-267, doi:10.1038/s41586-018-0753-3.
  23. Viukov, S.; Shani, T.; Bayerl, J.; Aguilera-Castrejon, A.; Oldak, B.; Sheban, D.; Tarazi, S.; Stelzer, Y.; Hanna, J.H.; Novershtern, N. Human primed and naive PSCs are both able to differentiate into trophoblast stem cells. Stem Cell Reports 2022, 17, 2484-2500, doi:10.1016/j.stemcr.2022.09.008.
  24. Wei, Y.; Wang, T.; Ma, L.; Zhang, Y.; Zhao, Y.; Lye, K.; Xiao, L.; Chen, C.; Wang, Z.; Ma, Y., et al. Efficient derivation of human trophoblast stem cells from primed pluripotent stem cells. Sci Adv 2021, 7, doi:10.1126/sciadv.abf4416.
  25. Wei, Y.; Zhou, X.; Huang, W.; Long, P.; Xiao, L.; Zhang, T.; Zhong, M.; Pan, G.; Ma, Y.; Yu, Y. Generation of trophoblast-like cells from the amnion in vitro: A novel cellular model for trophoblast development. Placenta 2017, 51, 28-37, doi:10.1016/j.placenta.2017.01.121.
  26. Yabe, S.; Alexenko, A.P.; Amita, M.; Yang, Y.; Schust, D.J.; Sadovsky, Y.; Ezashi, T.; Roberts, R.M. Comparison of syncytiotrophoblast generated from human embryonic stem cells and from term placentas. Proc Natl Acad Sci U S A 2016, 113, E2598-2607, doi:10.1073/pnas.1601630113.
  27. Yang, Y.; Adachi, K.; Sheridan, M.A.; Alexenko, A.P.; Schust, D.J.; Schulz, L.C.; Ezashi, T.; Roberts, R.M. Heightened potency of human pluripotent stem cell lines created by transient BMP4 exposure. Proc Natl Acad Sci U S A 2015, 112, E2337-2346, doi:10.1073/pnas.1504778112.
  28. Polakova, K.; Kuba, D.; Russ, G. The 4H84 monoclonal antibody detecting beta2m free nonclassical HLA-G molecules also binds to free heavy chains of classical HLA class I antigens present on activated lymphocytes. Hum Immunol 2004, 65, 157-162, doi:10.1016/j.humimm.2003.10.005.

Reviewer 2 Report

The manuscript's authors entitled "An improved protocol for targeted differentiation of human iP-2 SCs into HLA-G expressing trophoblasts cells enabling the 3 modelling of placenta-related disorders" intend to describe an efficient protocol for human iPS differentiation in trophoblasts.

1. Erase the dot in the title.

2. The figures have low resolution, and some writing can't be read.

The paper opens a perspective on optimizing differentiation protocols. It would be very interesting to see the promoter demethylation of the other genes and the expression of miR-103a target genes, both involved in trophoblast differentiation.

Author Response

Reviewer comment: The manuscript's authors entitled "An improved protocol for targeted differentiation of human iP-2 SCs into HLA-G expressing trophoblasts cells enabling the 3 modelling of placenta-related disorders" intend to describe an efficient protocol for human iPS differentiation in trophoblasts.

  1. Erase the dot in the title.

Done.

  1. The figures have low resolution, and some writing can't be read.

We revised the figures as requested.

The paper opens a perspective on optimizing differentiation protocols. It would be very interesting to see the promoter demethylation of the other genes and the expression of miR-103a target genes, both involved in trophoblast differentiation.

Thanks for the comment. Of course further characterization of promoter demethylation would be interesting as well as a deeper insight into target gene expression. These further analyses are out of the scope in this manuscript but are already under investigation for another project.

Round 2

Reviewer 1 Report

General remarks: Thanks for clarification of the strength of your study. Nevertheless, their are cell culture models using trophoblastic stem cells (without iPSCs generation) as mentioned by multiple answers of the author (available by isolation, commercially available (some regions) or easily shared by the scientific community), which have a one step protocol and a EVT generation ratio of 86% - 96.9% in recent publications. Therefore, an iPSCs model using a very expensive Cytotune® -iPS Sendai Reprogramming Kit (Thermo Fisher Scientific) technique, which is associated with a huge effort to characterize the resulting cell lines, is in our opinion not a significant improvement. Also the above-mentioned system is easily transferable into organoid systems and can be modified to mimic PE and other pregnancy-associated diseases. In conclusion, the present article by On-Chung Shum et al. shows a nicely improved protocol one step protocol for a "niche" application.

Additional comments: Thanks for adding the immune-lateral flow assay test to verify the hCG secretion, but as stated by the authors the primary objective of the study was focused on EVTs (HLAGpos cells). Therefore, the authors should consider to functionally verify this cell type with additional assays. Their data suggest the RNA upregulation of important marker protein and the expression of markers like HLAG, but as experienced scientists in stem cell research, they are likely aware that not all differentiated iPSCs cells are fully functional even with the correct marker subset.